# Comparison of SYK Signaling Networks Reveals the Potential Molecular Determinants of Its Tumor-Promoting and Suppressing Functions

**DOI:** 10.3390/biom11020308

**Published:** 2021-02-18

**Authors:** Marion Buffard, Aurélien Naldi, Gilles Freiss, Marcel Deckert, Ovidiu Radulescu, Peter J. Coopman, Romain M. Larive

**Affiliations:** 1IRCM, Université de Montpellier, ICM, INSERM, F-34298 Montpellier, France; marion.buffard75@gmail.com (M.B.); gilles.freiss@inserm.fr (G.F.); peter.coopman@inserm.fr (P.J.C.); 2LPHI, Université de Montpellier, CNRS, F-34095 Montpellier, France; ovidiu.radulescu@umontpellier.fr; 3Institut de Biologie de l'ENS (IBENS), Département de Biologie, École Normale Supérieure, CNRS, INSERM, Université PSL, F-75005 Paris, France; aurelien.naldi@gmail.com; 4Lifeware Group, Inria Saclay-île de France, F-91120 Palaiseau, France; 5C3M, Université Côte d'Azur, INSERM, équipe «Microenvironnement, Signalisation et Cancer», F-06204 Nice, France; Marcel.Deckert@unice.fr; 6CNRS—Centre National de la Recherche Scientifique, 1919 Route de Mende, F-34293 Montpellier, France; 7IBMM, Université Montpellier, CNRS, ENSCM, F-34093 Montpellier, France

**Keywords:** phosphoproteomics, SYK signaling, breast cancer, Burkitt lymphoma, Network comparison, oncogene, tumor suppressor

## Abstract

Spleen tyrosine kinase (SYK) can behave as an oncogene or a tumor suppressor, depending on the cell and tissue type. As pharmacological SYK inhibitors are currently evaluated in clinical trials, it is important to gain more information on the molecular mechanisms underpinning these opposite roles. To this aim, we reconstructed and compared its signaling networks using phosphoproteomic data from breast cancer and Burkitt lymphoma cell lines where SYK behaves as a tumor suppressor and promoter. Bioinformatic analyses allowed for unveiling the main differences in signaling pathways, network topology and signal propagation from SYK to its potential effectors. In breast cancer cells, the SYK target-enriched signaling pathways included intercellular adhesion and Hippo signaling components that are often linked to tumor suppression. In Burkitt lymphoma cells, the SYK target-enriched signaling pathways included molecules that could play a role in SYK pro-oncogenic function in B-cell lymphomas. Several protein interactions were profoundly rewired in the breast cancer network compared with the Burkitt lymphoma network. These data demonstrate that proteomic profiling combined with mathematical network modeling allows untangling complex pathway interplays and revealing difficult to discern interactions among the SYK pathways that positively and negatively affect tumor formation and progression.

## 1. Introduction

Protein tyrosine kinases, which account for 17% of the human kinome and are involved in the regulation of cell growth and invasion, are frequently associated with tumor formation and progression [1]. Therefore, they are interesting druggable targets [2].

In hematopoietic malignancies, the non-receptor spleen tyrosine kinase (SYK) mainly contributes to cancer cell survival and proliferation [3,4,5] by mediating tonic and chronic signaling through B-cell antigen receptors in different B-cell-derived lymphoma types [6,7,8]. On the other hand, we and others demonstrated that SYK is also expressed in non-hematopoietic cells and that it behaves as a tumor suppressor in mammary epithelial cells [9,10,11,12]. SYK anti-oncogenic and anti-invasive activities have been observed using mouse xenograft models of breast carcinoma [10], pancreatic carcinoma [13] and melanoma [14].

To better understand the apparent opposite functions of SYK depending on the cancer type (reviewed by [15,16]), we and other groups used quantitative phosphoproteomic approaches to identify novel SYK signaling effectors in breast cancer cells [17,18,19]. Moreover, we recently developed a bioinformatics pipeline that allows the reconstruction and analysis of the SYK signaling network from these global phosphoproteomic data [20]. This pipeline allows for identifying the intermediary proteins that propagate the signal from SYK to its effectors, instead of merely establishing a comprehensive list of direct and indirect SYK targets. We completed this tool with a graphical interface, validated using phosphoproteomic data on the phosphatidylinositol 4,5-bisphosphate 3-kinase catalytic subunit alpha (PIK3CA) and SRMS kinases [21].

Here, we used this pipeline to identify the molecular components responsible for SYK tumor-promoting and -suppressing functions by reconstructing the SYK signaling networks in Burkitt lymphoma cells (DG75 cells, data from [19,22]) and in MCF7 and MDA-MB-231 breast cancer cells. We then compared these networks to identify the similarities and differences that could help to explain SYK dual role in tumorigenesis.

## 2. Materials and Methods

### 2.1. Phosphoproteomic Data for the Bioinformatic Workflow

The bioinformatic workflow input consisted of a list of UniProt accession numbers (AC) of proteins identified as differentially phosphorylated or enriched (i.e., targets) between experimental conditions that lead to SYK perturbation. For MCF7 [17] and DG75 [19] (SYK-positive) cell phosphoproteomics, SYK was inhibited by incubating cells with its pharmacological inhibitor piceatannol or vehicle. For the SYK-negative MDA-MB-231 cell line (called “MDA231” in this study), we compared the phosphoproteomes of parental and inducible SYK-EGFP-expressing cells [18]. For DG75 and MDA231 cell phosphoproteomics, the phosphopeptides were obtained after trypsin-digestion prior to two steps purification on an anti-phospho-tyrosine affinity column and on polymer-based metal ion affinity column as previously described [18,19]. For MCF7 cell phosphoproteomics, the phosphoproteins were obtained after purification on an anti-phospho-tyrosine affinity column as previously described [17]. Thus, the MCF7 dataset also included the proteins that physically interact with phosphorylated proteins in a SYK-dependent manner.

To normalize the datasets, we selected only proteins the phosphorylation level of which was increased in the SYK-positive condition to allow the comparison with the DG75 dataset that contains only positive targets. In agreement with Naldi and colleagues [20], we added a posteriori the direct SYK phosphorylation of ezrin in the MCF7 and MDA231 arrays, and of E-cadherin and alpha-catenin in the MCF7 array. The complete list of SYK targets used as input is provided in Appendix A.

### 2.2. Online Databases

UniProt AC mapping from uniprot.org/downloads (accessed on 18 February 2021) (2017/02)

HGNC dataset from genenames.org/cgi-bin/statistics (accessed on 18 February 2021) (2017/02)

Gene ontology (GO) from geneontology.org/page/download-ontology (accessed on 18 February 2021) (go-basic.obo, 2017/02)

GO annotation from geneontology.org/page/download-annotations (accessed on 18 February 2021) (goa_human.gaf, 2017/02)

KEGG (Kyoto Encyclopedia of Genes and Genomes): www.kegg.jp (accessed on 18 February 2021), release 84 (2017/10)

Pathway Commons: pathwaycommons.org (accessed on 18 February 2021) release 8 (2018/01)

### 2.3. Gene Ontology Annotation

We annotated as phospho-tyrosine modifiers the components of the network with the tyrosine kinase (GO:0004713) and tyrosine phosphatase (GO:0004725) GO terms and extended the list of phospho-tyrosine modifiers from 123 proteins to 207 manually-verified proteins [21]. For subnetwork extraction, we used selected groups of GO terms that represent specific processes and functions, such as cell adhesion and motility (GO:0048870, GO:0007155, GO:0034330, GO:0022610, GO:0060352, GO:0030030), cell growth and death (GO:0008283, GO:0007049, GO:0008219, GO:0019835, GO:0000920, GO:0007569, GO:0051301, GO:0060242), immunity and inflammation (GO:0002376, GO:0001906), and cell differentiation (GO:0030154, GO:0036166) [20].

### 2.4. Pathway Database Selection

We used pathways from the KEGG [23,24,25] and Pathway Commons [26] databases. For signaling network reconstruction, we first selected the most enriched pathways in the lists of targets (using the Fisher’s exact test) and included the pathways containing targets not covered by significantly overrepresented pathways from the same database [20].

### 2.5. Hierarchical Clustering

The enrichment score is the additive inverse of the logarithm of the p-value obtained with the Fisher’s exact test. We performed the bidirectional hierarchical clustering of the enrichment scores for the KEGG pathways that included the SYK targets identified in the three phosphoproteomic studies using the R software (version 3.6.0; [27]) with the Rcmdr library [28].

### 2.6. Signaling Network Comparison

We embedded the selected pathways in a prior-knowledge signaling network for each cell line, as previously described [20,21]. We then compared different topological parameters using the NetworkAnalyzer tool from Cytoscape (version 3.7.1) [27,28] (http://www.cytoscape.org/ (accessed on 18 February 2021)). We removed isolated nodes and we considered the networks as directed. The topological parameters have been described previously [29].

We used the Cytoscape DyNet plug-in to align and compare the SYK networks [30] with the following parameters: initial layout of Prefuse Force Directed Layout; networks treated as directed; find corresponding nodes by name (UniProt ID); find corresponding edges by their source/target (we selected minimal edge properties in order not to discriminate them according to their database of origin). We removed self-loops (60–70 per network) and we did not find duplicated edges. Proximal networks included the first neighbors of the node corresponding to the proteins of interest and the interactions between them.

The Cytoscape files corresponding to the network representations are available in an open-access searchable directory (10.5281/zenodo.4091051).

### 2.7. Signal Propagation Analysis

We used our recently developed Phos2Net software to extract the shortest path subnetworks for each cell line [21]. Reconstructing prior-knowledge networks solely from SYK target-enriched signaling pathways has the advantage of selecting paths with greater confidence. However, we found that this reduced the number of reachable targets and created a bias in the comparison of the extracted subnetworks. Therefore, to focus our study on the intrinsic experimental dataset parameters, we used the same prior-knowledge network to analyze the signal propagation from SYK to its targets. We chose the “All pathways” selection mode to extract them from the same prior-knowledge network reconstructed from the components of all the KEGG and Pathway Commons databases, irrespective of their enrichment.

## 3. Results

### 3.1. Comparison of the SYK Signaling Networks in Breast Cancer and Burkitt Lymphoma Cell Lines

To better understand the molecular determinants of the SYK kinase pro- and anti-tumor activities, we analyzed existing tyrosine phosphoproteomic datasets that identified SYK targets in different cell types. Specifically, we used data obtained from the human breast cancer cell lines MCF7 and MDA231, in which SYK acts as a tumor suppressor [17,18,19], and from the human Burkitt lymphoma cell line DG75, in which SYK plays a pro-oncogenic role [19,22]. After post-treatment procedures and normalization of the original phosphoproteomic data (detailed in the Section 2), we used our bioinformatics pipeline to select SYK target-enriched signaling pathways, reconstruct signaling networks from elements of these pathways and select relatively small subnetworks that connect SYK to its targets [20,21]. We compared the results obtained for the breast cancer and Burkitt lymphoma cell lines at different stages of this analysis (Figure 1).

To start exploring the datasets, we first identified SYK targets common to at least two cell lines. The majority of the targets were cell line-specific (Appendix A). The largest shared subgroup included 90 proteins identified in both MDA231 and DG75 cells. Approximately 50% of these 90 proteins contained the same phosphorylation site(s), suggesting a similar impact of SYK. We then focused on SYK targets specific to each cell line. We investigated whether they were enriched in specific GO biological processes, but none showed a statistically significant enrichment. We observed the same lack of enrichment in GO terms for all the other SYK target subgroups (i.e., common to two or three cell lines).

### 3.2. KEGG Pathways Are Differentially Enriched in SYK Targets in the Breast Cancer and Burkitt Lymphoma Cell Datasets

To thoroughly compare SYK signaling networks in these three cell types, we asked whether the identified SYK targets belonged to specific signaling pathways, using the KEGG pathway database [23,24,25] and the Fisher’s exact test to assess the enrichment and classify them. The two breast cancer cell lines MCF7 and MDA231 did not cluster, and the distance between cell lines was closer for the MCF7 and DG75 cell lines than for the MCF7 and MDA231 cell line (Appendix A). To find signaling pathways that were differentially enriched in SYK targets in the three cell lines, we sorted them in function of the mean of the enrichment score in the three cell lines and calculated the deviation from this mean value for each line (Appendix A). We manually reviewed the signaling pathways with the highest differential enrichments and selected those in which the SYK targets were relevant to the functionality of the signaling pathway. To facilitate viewing, we colored proteins of the KEGG signaling pathways according to the SYK target list(s) they belonged to (DG75, MCF7 and/or MDA231 cell dataset), and we included for each KEGG signaling pathway a table that summarizes the main data for each SYK target. These files are available in an open-access searchable directory (10.5281/zenodo.4091051).

The B-cell receptor signaling pathway was the most enriched amongst the SYK targets in the DG75 cell line (Figure 2 and Table 1). SYK targets identified in MCF7 or MDA231 cells were not specific to this pathway and were involved in many other signaling pathways unrelated to lymphocyte-specific immunoreceptors. Amongst the DG75 cell-specific SYK targets identified in the B-cell receptor signaling pathway, the Bruton’s tyrosine kinase (BTK) protein tyrosine kinase is essential for B-lymphocyte development, differentiation and signaling [31]. In DG75 cells, BTK is phosphorylated on the Y551 residue, indicative of its catalytic activity [32]. Other signaling pathways enriched in DG75 cell-specific SYK targets included T-cell receptor signaling (Appendix A), Fc gamma receptor-mediated signaling (Appendix A), phagocytosis (Appendix A), and Th1 and Th2 cell differentiation (Appendix A). These pathways seem to be specific of cell types distinct from a B cell line, but the proteins and the protein interactions of these pathways are shared by many cell types.

In MCF7 and MDA231 breast cancer cells, the SYK targets most commonly found enriched belonged to the actin cytoskeleton regulation (Figure 3 and Table 2) and focal adhesion signaling (Appendix A). In these KEGG pathways, the SYK targets were specific either to the MCF7 or to the MDA231 cell line. In both cell lines, other enriched SYK targets belonged to the mRNA transport (Appendix A) and surveillance pathways (Appendix A). Interestingly, two signaling pathways involved in intercellular adhesion were particularly enriched in SYK targets in MCF7 cells: the adherens junction (Appendix A) and tight junction (Appendix A) pathways. SYK targets from these two signaling pathways were also part of the Hippo signaling pathway (Appendix A). Notably, YAP1, a key component of this pathway, was identified as a SYK target in MDA231 cells [18]. The Hippo signaling pathway is involved in cell contact-mediated inhibition of proliferation, suggesting a possible link between loss of SYK expression during breast tumor progression, loss of intercellular adhesion, and loss of control of mammary epithelial cell proliferation.

### 3.3. The SYK Signaling Networks Display Similar Topological Parameters

To compare the SYK signaling networks in the three cell types, we selected the signaling pathways that were most enriched in SYK targets from each dataset. We combined their components to create the SYK networks specific to each cell line using our Phos2Net bioinformatics tool. In these networks, nodes represent the proteins, and edges the interactions between these proteins. The DG75, MCF7 and MDA231 cell networks were similar, with a central core characterized by high density of hyperconnected nodes (Appendix A). This feature was absent in a network generated from a dataset of proteins randomly selected (Appendix A). The random network has been generated from the UniProt database. In addition, the node connectivity distribution showed a better power law correlation in the DG75, MCF7 and MDA231 cell networks than in the random network (Appendix A). A node connectivity distribution that approximates the power law distribution is called scale-free, and this network property is used to distinguish between random and scale-free network topologies [33,34,35]. Moreover, the length of the paths linking the proteins in the network was longer in the random network (Appendix A) and the path length distribution was shifted to longer paths in the random network compared with the DG75, MCF7 and MDA231 cell networks (Appendix A). Thus, the DG75, MCF7 and MDA231 cell networks generated from experimental data better matched the small-world properties characteristic of biological networks (i.e., high clustering coefficient and small path length) [36].

Next, we compared the topological parameters of the SYK nodes in the different networks (Appendix A). The only notable difference was the smaller number of direct connections in the MDA231 network than in the DG75 and MCF7 networks. This can be explained by the fact that for the phosphoproteomic analysis, the SYK-negative MDA231 cells were transfected to express exogenous SYK for the phosphoproteomic analysis [18]. Similarly, in the random network, SYK had fewer direct connections, and the connectivity of its neighbors was lower than in the other networks. The average path lengths through SYK were shorter in the DG75, MCF7 and MDA231 cell networks than in the random network. The closeness centrality (a metric that measures the average distance of a node to all other nodes) and betweenness centrality (that captures how much a given node is in-between others) values were lower in the random network. This suggests that SYK plays a role in rapid information control in the DG75, MCF7 and MDA231 cell networks [37,38].

Taken together, these results suggest that the topological parameters of the reconstructed networks are dataset-dependent and independent of our bioinformatics method.

### 3.4. SYK Targets Are Rewired between the Breast Cancer and Burkitt Lymphoma Cell Networks

To highlight the discriminating differences of the three SYK networks, we aligned the networks built using the DG75, MCF7 and MDA231 cell datasets. First, we merged the MCF7 and MDA231 cell networks, conserving the information on the origin of nodes and edges. We then compared this breast cancer SYK network with the Burkitt lymphoma SYK network. We defined the nodes and edges present in all networks as “common”, those present only in the DG75 network as “DG75-exclusive'', and those present only in the MCF7 and/or MDA231 cell networks as “breast cancer-exclusive” (Figure 4A,B).

We initially focused on the SYK proximal network by selecting its 68 first neighbors (Figure 4D). Almost all nodes were present in the three networks (but for 11 nodes absent from the MDA231 cell network). This subnetwork was hyper-connected with 736 edges among which 177 were DG75-exclusive and 94 breast cancer-exclusive. This last group included 29 edges shared by the MCF7 and MDA231 networks, 45 MCF7-exclusive edges, and 20 MDA231-exclusive edges. Among the 47 edges arriving at SYK, 12 were DG75-exclusive edges and corresponded to the protein interactions involved in SYK activation in lymphocyte signaling. The proteins involved in these interactions are generally not expressed in breast cancer cells. Moreover, there was no breast cancer-exclusive SYK edge, most probably due to the lack of information in the pathway databases on SYK activation in breast cancer cells.

We then selected amongst the identified SYK targets those that showed a major reconfiguration of their interactions in the breast cancer and Burkitt lymphoma SYK networks (Figure 4C). This parameter can be analyzed by quantifying the rewiring of the nodes corresponding to the SYK targets. We selected the most rewired SYK targets (Table 3; the complete list is available in Appendix A). The BLK protein tyrosine kinase is a member of the Src kinase family and is involved in immunoreceptor activation [39]. The BLK proximal network was mainly specific to the SYK signaling network in DG75 cells (Appendix A).

CBL is an E3 ubiquitin-protein ligase that functions as a negative regulator of many signaling pathways that are triggered by activation of cell surface receptors [40,41]. Interestingly, CBL interacts with SYK, and is phosphorylated in a SYK-dependent manner [42]. Conversely, SYK is negatively regulated by CBL and is a target of CBL-mediated ubiquitylation [43]. We found that CBL interactions were predominantly distributed in two distinct subsets of breast cancer and DG75 signaling pathways (Figure 4E). Although none of these proteins was DG75-exclusive, a significant part of the interactions were specific for the B-cell receptor activation signaling pathway. Some of the proteins were breast cancer-exclusive. Specific breast cancer interactions involved, for instance, the focal adhesion, DAP12 and PTK6 signaling pathways. Thus, our network analysis allowed identifying SYK targets that were specifically linked to breast cancer or Burkitt lymphoma specific subnetworks.

Two other rewired proteins were SKP1 and CDK1. SKP1 is part of the SKP, Cullin, F-box (SCF) E3 ubiquitin-protein ligase complex that is involved in the degradation of proteins implicated in the control of cell cycle progression and localized in the centrosome [44]. The serine/threonine kinase CDK1 plays a key role in cell cycle control [45,46,47]. Although we identified these proteins as SYK targets in two different datasets, we merged their proximal networks (Figure 4F) because they are both involved in cell cycle control, as reported for SYK [48,49,50]. Almost all SKP1 and CDK1 interactors were present in the breast cancer and Burkitt lymphoma networks. However, almost all interactions of CDK1 and those of SKP1 with CDK1 neighboring proteins were breast cancer-exclusive. Nevertheless, some interactions of CDK1 and SKP1 were DG75-exclusive, notably with identified SYK targets (VCPIP1 for CDK1, and COPS8 and COPS4 for SKP1). Although it was composed of target proteins shared by the MCF7 and DG75 cell datasets, the CDK1/SKP1 proximal network had mostly interactions that were exclusive to the breast cancer or DG75 cell network, suggesting that its regulation by SYK could be different depending on the cell type.

Finally, we were interested in the regulation of the Hippo signaling pathway because we observed that this pathway was enriched in MCF7 cell SYK targets and in YAP1 (MDA231 cell SYK target) (Appendix A). We included also the proximal STK4 network, another key component of the Hippo pathway [51], identified as an SYK target in the DG75 and MDA231 cell datasets. Half of the proteins were common to all three networks, and the other half were breast cancer-exclusive (Figure 4G). Almost all interactions were breast cancer-exclusive, whereas two STK4 interactions originating from the Ras signaling pathway were DG75-exclusive. It should be noted that the other identified SYK targets (AMOTL2, PPP2R1A and PPP2CA) came from the MCF7 cell dataset. These observations strengthen our hypothesis that SYK regulates the Hippo signaling pathway specifically in breast cancer cells.

### 3.5. Signal Propagation from SYK to its Targets Identified in the Breast Cancer and Burkitt Lymphoma Cell Lines

To connect SYK to its targets, we extracted the shortest paths between them for each cell line using our bioinformatics pipeline that combines the advantages of the random walk-based analysis methods and the weighted k-shortest paths [20,21]. We merged and colored the resulting subnetworks to compare them in the three cell lines.

The first SYK neighbors were 22 proteins with an interaction with SYK documented in signaling pathway databases and that link SYK to experimentally identified targets (Figure 5A). Among these 22 proteins, 16 were exclusive to only one cell line. Overall, the SYK subnetwork of the shortest pathways showed that many of them were exclusive to only one cell line (Figure 5B). This observation was confirmed when we restrained the paths to the targets involved in cell adhesion and motility, cell growth and death, differentiation, inflammation and transport and metabolism (Appendix A).

We observed that the Hippo signaling pathway was enriched in SYK targets in MCF7 cells (Appendix A), and that most interactions of the proximal network of YAP1 and STK4 were exclusive to the MCF7 and MDA231 cell lines (Figure 4G). We then selected paths leading to these two targets (Figure 5C). The shortest path to link SYK to STK4 involved the PLCG2 phospholipase, the HRAS GTPase and its effector RASSF5. We enlarged the selection to the near shortest paths that involve the phosphatidyl-inositol 3-kinase subunits (PIK3R5, PIK3R3 and PIK3CA) and caspase-3. However, as these pathways did not allow an explanation of STK4 phosphorylation, we cannot exclude the direct involvement of SYK or of another tyrosine kinase. For YAP1, the path from SYK ended with its phosphorylation by ABL1 that plays a key role in the response to DNA damage [52]. The first step of this pathway involved the FCER1G adaptor protein that allows linking integrin receptor engagement to SYK activation [53]. In addition, SYK can be activated in epithelial cells following integrin engagement [54]. However, in a classical activation cascade, FCER1G should be situated upstream of SYK. The involvement of other proximal SYK effectors in YAP1 regulation needs to be investigated to clarify this pathway.

## 4. Discussion

Elucidating the molecular determinants that explain the causes and consequences of deregulated signaling pathways is a major goal in cancer cell biology. The current advances in proteomic profiling allow for unraveling complex signaling pathways. Network analysis permits us to deal with the complex interplay between these pathways and to reveal the difficult-to-discern crosstalk between the pathways that positively and negatively affect tumor growth, progression and drug resistance.

The SYK kinase plays a dual role as tumor promoter and tumor suppressor [16]. To better understand these seemingly contradictory functions, we compared SYK phosphoproteomic data from breast carcinoma and Burkitt lymphoma cells using our in-house Phos2Net bioinformatics tool that reconstructs signaling networks from phosphoproteomic data. This allowed us to demonstrate that, in breast cancer cells, the SYK target-enriched signaling pathways included intercellular adhesion and Hippo signaling components that are often linked to tumor suppression. In Burkitt lymphoma cells, the SYK target-enriched signaling pathways included molecules that could play a role in SYK pro-oncogenic function in B-cell lymphomas. Several protein interactions were profoundly rewired in the breast cancer network compared with the Burkitt lymphoma network. SYK targets that are common to these different cell types, such as the CBL, CDK1 and SKP1 proteins, are differentially affected depending on the cellular context.

We found that two different sets of signaling pathways are enriched in SYK targets in breast cancer cells and Burkitt lymphoma cells (Figure 6). For instance, SYK ability to regulate the actin cytoskeleton and focal adhesion signaling is much stronger in breast cancer cells. Indeed, SYK relocates to the actin filament network and subsequently associates with focal adhesion kinase (FAK) [55]. We demonstrated that SYK regulates cortactin and ezrin, both involved in actin-mediated cell adhesion and motility [20], and that SYK stimulates the interaction/stability of the E-cadherin/catenin complex [56]. The mRNA transport and surveillance pathways also were also particularly enriched in SYK targets in breast cancer cells. There is almost no information on SYK and mRNA, except for its role in stabilization of the mRNA encoding the BCL-XL protein, thereby protecting cells from apoptosis induced by oxidative or genotoxic stress [57]. Adherens and tight junctions seem to be specific SYK targets in MCF7 cells, which confirms observations by us and other groups on the regulation of the E-cadherin/catenin pathway by SYK [17,56,58,59]. Furthermore, our observations indicate a possible link between loss of SYK expression during breast tumor progression, loss of intercellular adhesion, and loss of control of mammary epithelial cell proliferation via the Hippo signaling pathway. Particularly, in the reconstructed SYK signaling network, the proximal network of YAP1 and STK4, two Hippo pathway proteins, is almost entirely exclusive to breast cancer cells. However, it should be noted that the Hippo kinase STK4, also identified as a SYK target in DG75 cells, and YAP1, are involved in hematologic malignancies [60,61]. A very recent study reported that ezrin phosphorylation via SYK is associated with the inactivation of the Hippo pathway and an increase in YAP1 expression [62]. Tyrosine phosphorylation of YAP1 and STK4 induce pro- or anti-tumoral effects, depending on the cellular context [52,63,64,65,66]. Thus, the Hippo/YAP1 association with SYK needs to be investigated to clarify how it might correlate with its tumor suppressing function.

In Burkitt lymphoma cells, the B-cell receptor signaling pathway was the most enriched in SYK targets, including the BTK protein tyrosine kinase. Its oncogenic function is dependent on the cell developmental stage. Indeed, it is a tumor suppressor in pre-B cells, and a tumor-promoting factor in mature B cells [67]. DG75 cells are lymphoblastoid B cells, derived from a metastatic Burkitt lymphoma [68]. Therefore, BTK could be crucial for SYK pro-oncogenic function in B-cell lymphomas. We also found other pathways downstream of immunoreceptors involved in lymphocyte and myelocyte lineage signaling, receptors that are not expressed in epithelial cells. Consistent with this observation, we observed that the BLK protein tyrosine kinase proximal network was specific to the SYK signaling network in DG75 cells.

The alignment of the networks reconstructed from the different datasets allowed us to demonstrate that protein interactions of some SYK targets common to the networks were profoundly modified between the breast cancer and lymphoma networks. The paradigmatic example is the E3 ubiquitin-protein ligase CBL that is expressed in both hematopoietic and epithelial cells. CBL has been identified as a proto-oncogene in hematopoietic cells [69]. Conversely, CBL expression in the tumor positively correlates with survival in patients with metastatic colorectal cancer [70], and its CBLB homologue predicts better prognosis in breast cancer [71]. Therefore, CBL may also exert a dual function in cancer, depending on the cellular context [41].

Interestingly, we observed that the CDK1 and SKP1 proteins, identified as SYK targets in DG75 and MCF7 cells, respectively, are part of a network of interactions specific to each cell type. CDK1 is the master kinase that controls DNA replication and mitotic entry and is negatively regulated by tyrosine phosphorylation [72,73]. SKP1 is an adaptor component of the SCF E3 ubiquitin ligase complex that exerts an oncogenic function in cancer and that targets regulators of cell cycle progression. SKP1 is a target of anti-cancer therapeutic interventions [44], but its regulation by phosphorylation remains unknown.

An in-depth comparison of the SYK interactions that were rewired between the breast cancer and Burkitt lymphoma networks revealed the absence of breast cancer cell-specific interactions arriving at SYK. This is a consequence of the absence of defined SYK activation pathways in non-hematopoietic cells in the databases. SYK can be activated in epithelial cells by the engagement of beta1 integrin receptors, through reoxygenation and tumor necrosis factor alpha [54,74,75], but the activation mechanism has not been clearly established yet. The identification of SYK activation pathways in mammary epithelial cells would allow completing the signaling network and better understanding its breast tumor suppressor function.

Overall, a better characterization of the SYK phosphorylation signaling pathways in different tissue types is crucial because pharmacological SYK inhibitors are now used in the clinic to treat immune thrombocytopenia, autoimmune hemolytic anemia and IgA nephropathy [76]. Indeed, Rivera and Colbert wrote that “the long-term use of SYK inhibitors should be closely monitored and its use might be inappropriate for people with a family history of breast cancer” [77], whereas Khan and coworkers asked whether “oral spleen tyrosine kinase inhibitors could lead to neoplastic transformation” [78]. With the advent of precision medicine, it is clear that protein networks are more robust biomarkers than individual proteins and genes. Drug development is shifting from individual gene to dynamic network targeting by time- and order-dependent drug combinations. Targeting SYK effectors might be more efficient and specific than the global inhibition of its kinase activity. In conclusion, this manuscript brings insights into the cell type-specific SYK targets and unveils their signaling networks in breast cancer and Burkitt lymphoma cells. It demonstrates that although some of the SYK targets are shared by many cancer cell types, their protein interactions may be different and thus have a potentially different impact on cell behavior.

## Figures and Tables

**Figure 1 biomolecules-11-00308-f001:**
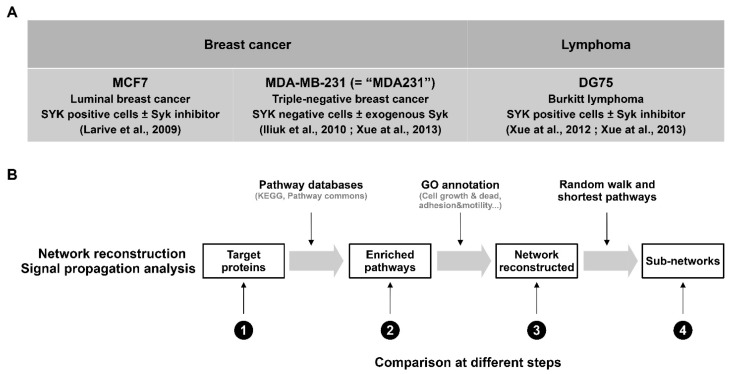
Schematic representation of the approaches used to analyze and compare SYK signaling in different cancer cell lines. (**A**) Cell lines and the corresponding phosphoproteomic analysis strategies used by the authors of the quoted studies to identify SYK targets. We analyzed these phosphoproteomic data to bootstrap the reconstruction of comprehensive networks of SYK downstream signaling in breast cancer and lymphoma cells. (**B**) Workflow of the network reconstruction and signal propagation analysis (according to [20,21]) and comparison of the results obtained in the three cell lines at different steps of this pipeline.

**Figure 2 biomolecules-11-00308-f002:**
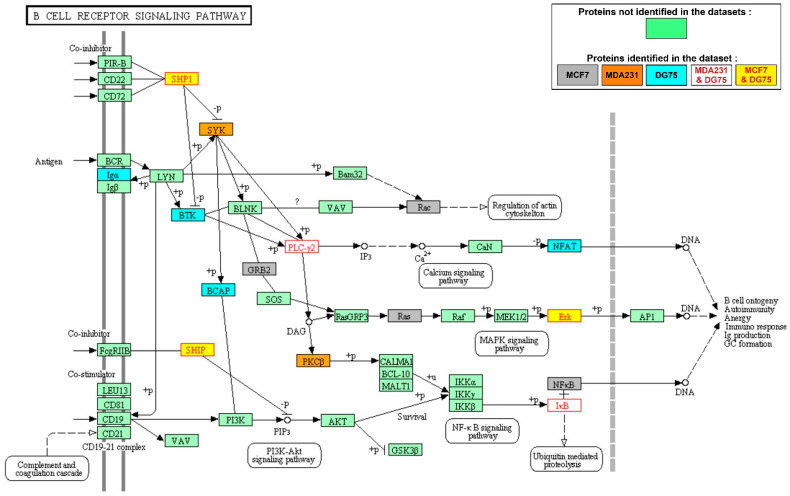
The B-cell receptor signaling pathway is enriched in SYK targets in the DG75 Burkitt lymphoma cell line. Representation of the KEGG signaling pathway; SYK targets from the different datasets are in different colors, as indicated in the box (step 2 of the workflow described in Figure 1B).

**Figure 3 biomolecules-11-00308-f003:**
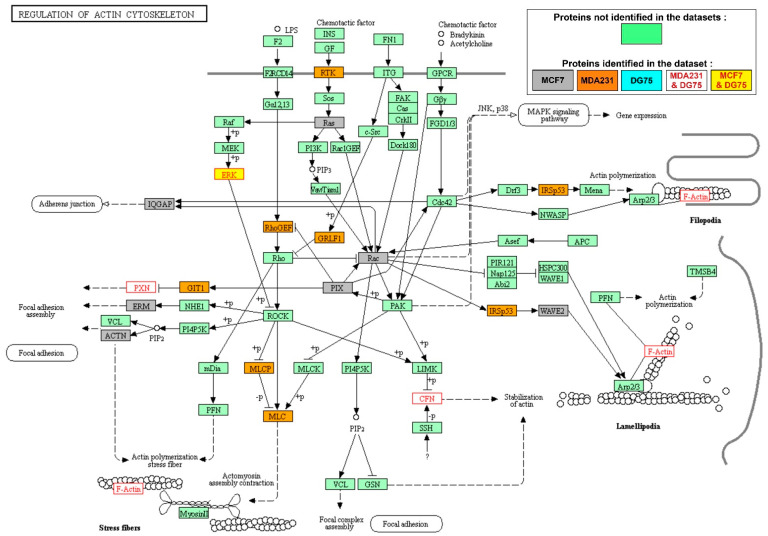
The actin cytoskeleton regulation pathway is enriched in SYK targets in the MCF7 and MDA231 breast cancer cell lines. Representation of the KEGG signaling pathway; SYK targets from the different datasets are in different colors, as indicated in the box (step 2 of the workflow described in Figure 1B).

**Figure 4 biomolecules-11-00308-f004:**
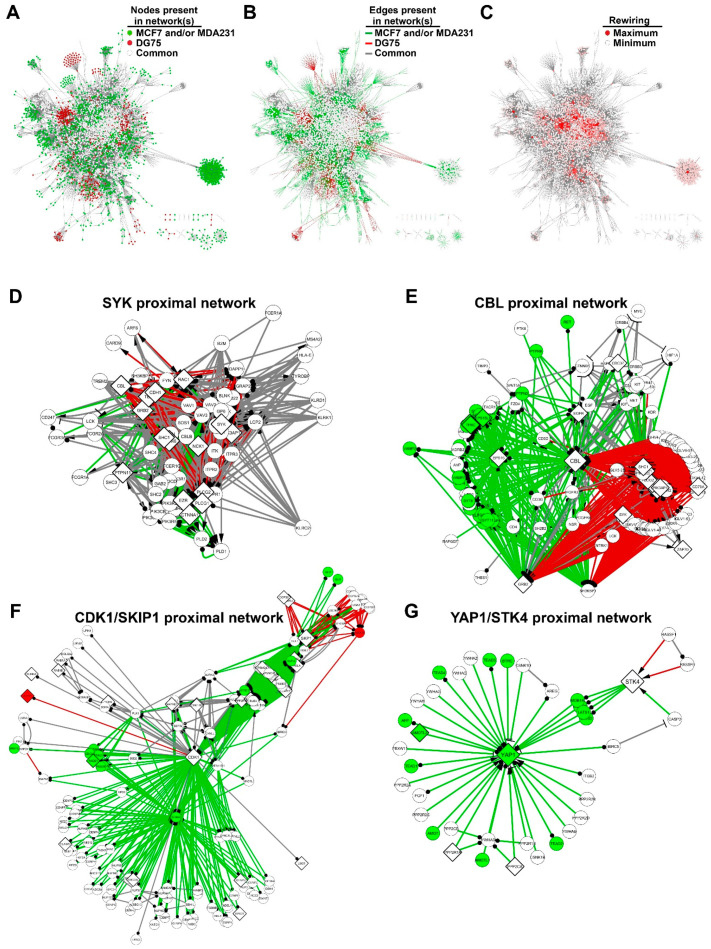
Alignment of the networks allows identifying SYK targets that are differentially rewired in the breast cancer and Burkitt lymphoma networks. Networks are obtained by combining the selected pathways (step 2 of the workflow described in Figure 1B). Nodes with a diamond or circle shape correspond to experimentally identified targets or to proTable 75. cell networks. The MCF7 and MDA231 cell networks were first aligned to generate the breast cancer network. (**A**) The nodes of the fusion networks are colored depending on their exclusive presence in the MCF7 and/or MDA231 cell networks (green), in the DG75 cell network (red), or in all three networks (white). (**B**) The edges of the fusion networks are colored depending on their exclusive presence in the MCF7 and/or MDA231 cell networks (green), in the DG75 cell network (red), or in all three networks (gray). (**C**) The nodes of the fusion network are colored using a red intensity scale depending on the rewiring of their connections between the MCF7/MDA231 and DG75 cell networks. (**D**–**G**) Subnetworks of the first SYK neighbors (**D**) and of the most rewired SYK targets (**E**–**G**). Nodes and edges are colored as described in (**A**,**B**). Diamond-shaped nodes are SYK targets. (**E**) Subnetwork of the first neighbors of CBL. (**F**) Subnetwork of the first neighbors of CDK1 and SKP1. (**G**) Subnetwork of the first neighbors of YAP1 and STK4.

**Figure 5 biomolecules-11-00308-f005:**
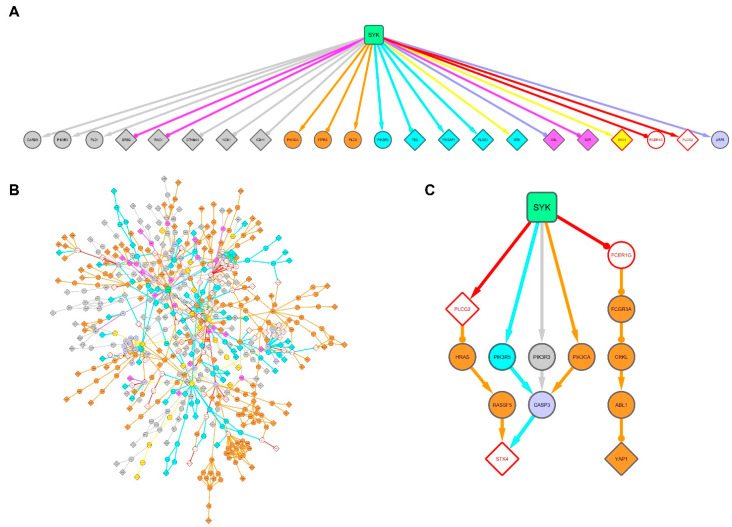
Signal propagation analysis from SYK to its targets identified in the breast cancer and the Burkitt lymphoma cell line. The paths from SYK to its targets identified in the MCF7, MDA231 and/or the DG75 cell datasets were extracted from the same prior-knowledge network (step 4 of the workflow in Figure 1B). Diamond-shaped nodes are SYK targets and are colored according to their presence in a specific experimental dataset; circle-shaped nodes are present in the prior-knowledge network and are colored according to their presence in the extracted paths (gray, MCF7; orange, MDA231; blue, DG75; pink, MCF7 and MDA231; yellow with red border, MCF7 and DG75; white with red border, MDA231 and DG75; purple, MCF7, MDA231 and DG75). Edges are colored depending on their presence in the extracted paths (gray, MCF7; orange, MDA231; blue, DG75; pink, MCF7 and MDA231; yellow, MCF7 and DG75; red, MDA231 and DG75; purple, MCF7, MDA231 and DG75). The arrow shape of the edges corresponds to the type of interaction (Delta, positive interaction; T, negative interaction; Circle, unknown). (**A**) SYK-specific subnetwork to its direct targets and first neighbors. (**B**) Subnetwork of the signal propagation from SYK to its targets. (**C**) Subnetwork specific of the signal propagation from SYK to YAP1 and STK4.

**Figure 6 biomolecules-11-00308-f006:**
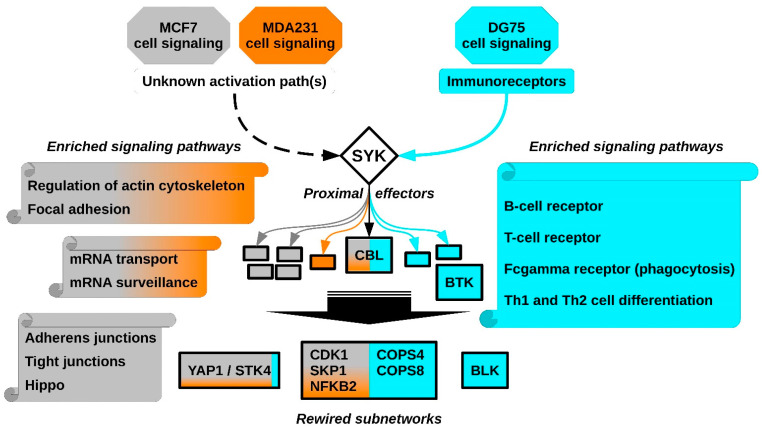
Schematic overview of the SYK signaling networks in breast cancer and Burkitt lymphoma cell lines.

**Table 1 biomolecules-11-00308-t001:** SYK targets from the different datasets present in the KEGG B-cell receptor signaling pathway.

Gene Symbol	KEGG Entry	UniProt Entry	MCF7 Cell Dataset (Present)	MDA231 Cell Dataset (Phosphosites)	DG75 Cell Dataset (Phosphosites)
NFKBIE	4794	O00221		155	155
NFATC2	4773	Q13469			752
PTPN6	5777	P29350	X		536
NRAS	4893	P01111	X		
RAC1	5879	P63000	X		
GRB2	2885	P62993	X		
PLCG2	5336	P16885		1217	759 * ; 550 ; 858 ; 1245
BTK	695	Q06187			551 *
MAPK3	5595	P27361			204
RELA	5970	Q04206	X		
SYK	6850	P43405		352	
INPPL1	3636	O15357	X		1135
PRKCB	5579	P05771		507	
PIK3AP1	118788	Q6ZUJ8			694
CD79A	973	P11912			210
MAPK1	5594	P28482	X		187

*, phosphosites that could be directly phosphorylated by SYK.

**Table 2 biomolecules-11-00308-t002:** SYK targets from the different datasets present in the KEGG Regulation of Actin Cytoskeleton pathway.

Gene Symbol	KEGG Entry	UniProt Entry	MCF7 Cell Dataset (Present)	MDA231 Cell Dataset (Phosphosites)	DG75 Cell Dataset (Phosphosites)
PXN	5829	P49023		409 ; 468	88
WASF2	10163	Q9Y6W5	X		
MYL12B	103910	O14950		143	
ARHGEF12	23365	Q9NZN5	X	1232	
NRAS	4893	P01111	X		
ACTN1	87	P12814	X		
RDX	5962	P35241			
PPP1R12A	4659	O14974		446 ; 446	
CFL2	1073	Q9Y281		89 *	89
ARHGEF7	8874	Q14155	X		
PPP1CB	5500	P62140	X		
MAPK1	5594	P28482	X		187
ACTB	60	P60709		240	240
CFL1	1072	P23528		89 *	68 ; 89
GIT1	28964	Q9Y2X7		598 ; 545	
PPP1CA	5499	P62136	X		
RAC1	5879	P63000	X		
EZR	7430	P15311	X	424	
IQGAP2	10788	Q13576	X		
MAPK3	5595	P27361			204
MYL12A	10627	P19105	X	142	
EGFR	1956	P00533		1069	
ACTN4	81	O43707	X		
ARHGAP35	2909	Q9NRY4		1087	
ACTG1	71	P63261		240	240
BAIAP2	10458	Q9UQB8		505	

*, phosphosites that could be directly phosphorylated by SYK.

**Table 3 biomolecules-11-00308-t003:** The most rewired SYK targets between the breast cancer cell networks and the lymphoma cell network.

Gene Symbol	UniProt Entry	Target in Network(s) [Site(s) of Phosphorylation]	Present in Network(s)
ARFGAP1	Q8N6T3	MCF7 ; MDA231 [Y175]	MCF7 ; MDA231
BLK	P51451	DG75 [Y187]	MCF7 ; DG75
CBL	P22681	MCF7 ; MDA231 [Y700]	MCF7 ; MDA231 ; DG75
CDK1	P06493	DG75 [Y15]	MCF7 ; MDA231 ; DG75
CLTA	P09496	MCF7	MCF7 ; MDA231 ; DG75
COPS4	Q9BT78	MCF7	MCF7 ; DG75
COPS8	Q99627	MCF7	MCF7 ; DG75
DNMT1	P26358	MDA231 [Y359 ; Y399]	MDA231
EPS15L1	Q9UBC2	MCF7	MCF7 ; MDA231
ESPL1	Q14674	MDA231 [Y719]	MCF7 ; MDA231 ; DG75
FIP1L1	Q6UN15	DG75 [Y113]	DG75
GBF1	Q92538	MDA231 [Y377]	MDA231
KPNA4	O00629	MDA231 [Y66]	MDA231
MTMR6	Q9Y217	MDA231 [Y108]	MDA231
NFKB2	Q00653	DG75 [Y39 ; Y867]	MCF7 ; MDA231 ; DG75
PTDSS1	P48651	MDA231 [Y424]	MDA231
PYGB	P11216	MDA231 [Y263]	MCF7 ; MDA231
REEP4	Q9H6H4	MDA231 [Y172]	MDA231
RRP8	O43159	DG75 [Y259]	DG75
SAP30BP	Q9UHR5	MDA231 [Y46]	MDA231
SF3B1	O75533	MDA231 [Y38 ; Y70]	MDA231
SKP1	P63208	MCF7	MCF7 ; DG75
STAM	Q92783	MCF7	MCF7 ; MDA231
SUPT5H	O00267	MDA231 [Y140]	MDA231
TFRC	P02786	MDA231 [Y20]	MCF7 ; MDA231
TRIP10	Q15642	MCF7	MCF7 ; DG75
TYMS	P04818	MDA231 [Y153]	MCF7 ; MDA231
UPF1	Q92900	MDA231 [Y113 ; Y1112] ; DG75 [Y113 ; Y1112]	MCF7 ; MDA231 ; DG75
VAMP8	Q9BV40	MCF7	MCF7 ; MDA231
YAP1	P46937	MDA231 [Y407 ; Y391]	MCF7 ; MDA231

## Data Availability

The Cytoscape files corresponding to the network representations and the KEGG signaling pathways colored according to the SYK target list(s) they belonged to (DG75, MCF7 and/or MDA231 cell dataset), with a table that summarizes the main data for each SYK target, are available in an open-access searchable directory (10.5281/zenodo.4091051).

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
