# Peer review of "Comparison of SYK Signaling Networks Reveals the Potential Molecular Determinants of Its Tumor-Promoting and Suppressing Functions"

_biomolecules, 2021, doi:10.3390/biom11020308_

Round 1
Reviewer 1 Report
The authors follow-up on a previous report in which they combined the shortest paths method with random walk processes to re-construct a regulatory network of Syk and its targets in breast cancer cell lines MCF7 and MBA231. To dissect the main differences of SYK signaling in tumor promoting and suppressing functions, Marion et al. used the tyrosine phosphoproteomic datasets from breast cell lines MCF7 and MBA231 and DG75 B cells to re-analyze the SYK target-enriched signaling pathway. They found that the SYK targets from the datasets of breast cancer cell lines and Burkitt lymphoma cell line were differentially enriched in KEGG signaling pathways: B-cell receptor signaling pathway was the most enriched in the DG75 cell dataset, whereas actin cytoskeleton regulation was the most targeted in MCF7 and MBA231 datasets. The authors further compared the common and cancer cell exclusive SYK target networks and identified several targets specifically linked to cancer specific subnetworks, including the Hippo signaling pathway in breast cancer cells. These data provided a better characterization of the SYK involved signaling pathway in different cancer cell types, however, many SYK targets and SYK involved regulatory networks were identified by the authors’ previous studies (for example, Naldi et al. (2017) PloS Comput Biol 13 (3): e1005432) and by the other groups, so the major novelty in the manuscript is not fully demonstrated. This should be clearly addressed further in the revised manuscript.
The following questions also should be addressed and/or discussed:
- Some of the figures in the manuscript is not in good quality and hard to read. For example, Figure 4; Fig5A.
- Table 3 is missing in the manuscript.
- Figure 6 and Figure S2: The author showed that the T-cell receptor signaling pathway is enriched in SYK targets from the DG75 Burkitt lymphoma cell line data, and Th1 and Th2 cell differentiation signal also enriched in this cell line dataset. However, DG75 is a B cell line, can the authors explain this in the manuscript? Is this result false positive?
- Some of the data in the supplemental table are not consistent with the published results. For example, in Table S5, the phosphosite of NFYA from the DG75 cell dataset is Y237 in the ref 71, but your dataset showed it is on 266; the phosphosite of PTPN11 is Y580, but your data is 584, etc. The authors should clarify these results.
Author Response
Please find attached the rebuttal letter with a point-by-point response to the issues raised by the reviewer 1.

Reviewer 2 Report
This study used a proteomics-based approach to investigate the opposing role of SYK signaling in breast cancer and Burkitt lymphoma cells. This research area (bioinformatics) is not the expertise of research of this reviewer, and thus this reviewer may not be in a position to evaluate the originality/novelty of this study. However their findings are significant and scientifically sound. Also, this study is timely since it is of great interest in cancer research area to study the dual/opposing role of the signaling proteins depending on the tissue type or context.
As an experimentalist, this reviewer may not be able to evaluate the main research efforts of the manuscript, thus limit comments to experimental component.
Comment: The authors used data of studies using Piceatannol as an SYK inhibitor. However, this drug is known to affect NF-kB signaling as well. Importantly, NF-kB also affects signaling associated with various cancers. Other data using more specific SYK inhibitor(s) could be included to validate this study.
Author Response
Please find attached the rebuttal letter with a point-by-point response to the issues raised by the reviewer 2.

Reviewer 3 Report
The manuscript "Comparison of SYK Signaling Networks Reveals the Potential Molecular Determinants of its Tumor-Promoting and -Suppressing Functions" provides an interesting and original outlook on what may drive polar SYK functioning in lymphoma and breast cancer. The work is overall thoroughly done, well written and structured. It is overall a nice example of creating new knowledge through reuse of previously reported phosphoproteomics data.
My specific comments and suggestions for this work would be:
- the authors should rephrase their description of the SYK involvement Hippo/YAP1 pathway to make it more precise. It is well-known (and cited in some of the author's own references) that inhibition of Hippo and/or activation of YAP1 has tumor-promoting properties in the breast cancers. It should be made more clear how the Hippo/YAP1 association of SYK might correlate with its tumor-suppressing function.
- similarly, overall authors address the main topic of the study in rather general terms. They should rephrase the Discussion to make more clear how their exact findings might answer the dual function of SYK in different cancer types.
- since the datasets were obtained by different labs and also using a somewhat different approach in one case, the authors should provide a detailed technical description of how this might affect the results.
- as an overall suggestion, authors might also consider additional weighing of the components of enriched pathways using the overall expression data from databases (e.g. mRNA or proteins from CCLE) to refine their networks. This might help to eliminate components with high score in terms of changing their phosphorylation state, but low expression and thus low likelihood that this change has significant effects in the signaling.
Some minor observations:
- Figure S1C should be modified to, e.g. barplot or a simple dot plot with no connecting lines - connecting the points of scores of unrelated pathways has no meaning and creates noise. Vertical lines might be also helpful.
- The proteins in table S1 are provided as Uniprot IDs, it would be nice to add the symbols/names as well to simplify understanding by readers.
Author Response
Please find attached the rebuttal letter with a point-by-point response to the issues raised by the reviewer 3.

Round 2
Reviewer 1 Report
The authors have addressed all my concerns. I do not have the other questions.